# Anterograde Neuronal Circuit Tracers Derived from Herpes Simplex Virus 1: Development, Application, and Perspectives

**DOI:** 10.3390/ijms21165937

**Published:** 2020-08-18

**Authors:** Dong Li, Hong Yang, Feng Xiong, Xiangmin Xu, Wen-Bo Zeng, Fei Zhao, Min-Hua Luo

**Affiliations:** 1State Key Laboratory of Virology, CAS Center for Excellence in Brain Science and Intelligence Technology, Center for Biosafety Mega-Science, Wuhan Institute of Virology, Chinese Academy of Sciences, Wuhan 430071, China; lidong@cibr.ac.cn (D.L.); yanghong_g@hotmail.com (H.Y.); xiongfengwhiov@hotmail.com (F.X.); zengwb@wh.iov.cn (W.-B.Z.); 2University of Chinese Academy of Sciences, Beijing 100049, China; 3Department of Anatomy and Neurobiology, School of Medicine, University of California, Irvine, CA 92697-1275, USA; xiangmix@hs.uci.edu; 4School of Basic Medical Sciences, Capital Medical University, Beijing 100069, China; 5Chinese Institute for Brain Research, Beijing 102206, China

**Keywords:** Herpes simplex virus 1, HSV-1, strain H129 (H129), viral tracer, neuroscience, neuronal circuit, anterograde, development, improvement, limitation

## Abstract

Herpes simplex virus type 1 (HSV-1) has great potential to be applied as a viral tool for gene delivery or oncolysis. The broad infection tropism of HSV-1 makes it a suitable tool for targeting many different cell types, and its 150 kb double-stranded DNA genome provides great capacity for exogenous genes. Moreover, the features of neuron infection and neuron-to-neuron spread also offer special value to neuroscience. HSV-1 strain H129, with its predominant anterograde transneuronal transmission, represents one of the most promising anterograde neuronal circuit tracers to map output neuronal pathways. Decades of development have greatly expanded the H129-derived anterograde tracing toolbox, including polysynaptic and monosynaptic tracers with various fluorescent protein labeling. These tracers have been applied to neuroanatomical studies, and have contributed to revealing multiple important neuronal circuits. However, current H129-derived tracers retain intrinsic drawbacks that limit their broad application, such as yet-to-be improved labeling intensity, potential nonspecific retrograde labeling, and high toxicity. The biological complexity of HSV-1 and its insufficiently characterized virological properties have caused difficulties in its improvement and optimization as a viral tool. In this review, we focus on the current H129-derived viral tracers and highlight strategies in which future technological development can advance its use as a tool.

## 1. Introduction

Herpes simplex virus 1 (HSV-1) belongs to the α-subfamily of *Herpesviridae*, which is distinguished from the β- and γ-subfamilies by its fast reproduction and the ability to infect and establish latency in neurons [1]. As a ubiquitous and highly contagious human pathogen, HSV-1 has gained very high infection prevalence worldwide. Sera-antibody-based epidemiological surveys show that 67% of the global population under the age of 50, equivalent to about 3.7 billion people, have been infected with HSV-1 [2]. In certain developing countries or regions, such as central and eastern China, HSV-1 seroprevalence reaches 90% [3,4]. Most of the HSV-1 infected population are asymptomatic, with cold sores being the most common clinical manifestation. HSV-1 also occasionally causes severe or even life-threatening diseases such as retinitis, keratitis [5], encephalitis [6], and systemic infection [7]. These lead to higher morbidity and mortality rates, especially in vulnerable neonates and immunosuppressed individuals [8]. Growing evidence supports the direct relationship between HSV-1 infection and Alzheimer’s disease [9,10], highlighting the important influence of this neurotropic virus on the nervous system.

The primary HSV-1 infection normally occurs during childhood and is mostly asymptomatic. The virus initially replicates in the mucosal epithelial cells by a typical expression cascade of immediate-early (IE), early (E), and late (L) genes. The propagated viruses may then invade the innervating sensory neurons via axonal termini and retrogradely transport to their soma in the trigeminal ganglion, where the viruses rapidly enter a quiescent state and establish a lifelong latency [11]. When the host encounters immunocompromised conditions due to psychological stress, disease, immunosuppressive regimen, or other causes, the latent HSV-1 may reactivate and resume productive replication again [12]. The reactivated viruses anterogradely transmit along the axon and infect peripheral tissues to cause recurrent infection, commonly known as cold sores. In some cases, the reactivated virus may invade the central nervous system by infecting further neuronal cells causing encephalitis. The ability to establish latency and the reactivating potential make HSV-1 extremely difficult to eliminate completely.

Despite being a hazardous pathogen, HSV-1 also has certain unique features that offer application potential. The ~150 kb double-stranded (ds) DNA genome, with about half of it being non-essential genes, provides plenty of capacity for exogenous genes, making HSV-1 an ideal vehicle for large gene delivery (reviewed in Reference [13]). HSV-1 has also been approved to be powerful oncolytic therapy for certain tumors [14,15].

Neurotropism and efficient infection of the central nervous system (CNS) make HSV-1 a candidate for being a neuron-specific labeling and gene delivery vector [16]. Moreover, the transneuronal transmission capacities of HSV-1 further highlight its potentiality to be employed for visualizing neuronal connections. HSV-1 strain H129 (H129), predominantly transmitting through neurons via an anterograde direction [17], represents one of the most promising viral tracers to map output neuronal circuits, and therefore has been genetically modified to develop neuronal circuit tracers [18]. Multiple neuroscience laboratories worldwide have applied H129-derived tools in their research and have achieved numerous important findings. However, current H129-derived tracers still have many drawbacks. Efforts from both virology and neurology fields are required to address these drawbacks and create improved H129-derived tracers.

## 2. Neuronal Circuit and Traditional Tracers

Neurons connect with each other through synapses, wiring to form complicated but precise networks. Information is processed by and transmitted among certain neurons, namely circuits, which are the basis of brain functions. Understanding the entire brain connectivity, also known as the connectome, by mapping neuronal circuits, is a vital task for neuroscience research. This will eventually contribute to revealing how the brain works. By deciphering the neuronal connections at the mesoscale level, the connectome plays an important role in bridging macroscale (whole brain or brain region level) and microscale (synapse or molecule level) information.

Mapping neuronal circuits requires proper tracers that can label the neurons in the initial area and then transmit to the upper or downstream connected areas. Traditional tracers include inorganic fluorescent molecules [19,20], lectins [21], bacterial toxins [22], autoradiographic amino acids [23], *etc* (reviewed in Reference [24]). But these traditional tracers are not the ideal solutions for circuit tracing because they have no, or only very low levels of, transneuronal transmission capability. The non-amplifiable labeling signal of these tracers also causes low labeling intensity or fast dilution/reduction during transmission. Therefore, traditional tracers cannot efficiently label neurons in the area of interest and map neuronal circuits. In addition, most of these tracers label neurons via passive uptake and show no cell-type specificity, resulting in low labeling fidelity and selectivity. These intrinsic limitations restrict the further application of these traditional tracers.

## 3. Viral Tracers

Neurotropic viruses provide more promising tracing tool candidates that overcome the drawbacks of traditional tracers. These microorganisms naturally invade and replicate in neurons, and then their progenies transmit to the connected or neighboring neurons. Benefiting from modern virology and genetic engineering techniques, viral tracers that have been derived from neurotropic viruses represent the most promising and powerful tracing tools in neuroscience research.

There are two different types of viral tracers distinguished by transmission capacity. Non-transmittable viral tracers, normally represented by adeno-associated virus (AAV), reside in the initially infected neurons and express the designated fluorescent protein or exogenous genes. Besides being applied as a gene delivery tool for genetic management and functional analysis, AAVs are also often used as neuronal circuit tracers to label neurons and their axons with fluorescent proteins. AAV tracers have been mostly applied to obtain information about innervation by highlighting axons, but had usually not been applied to obtain the true connection information, which requires transsynaptic transmission.

Transmittable viral tracers spread from one neuron to connected neurons. Along with transneuronal transmission, these viral tracers can be used to label infected neurons with fluorescent proteins. The direction along which the neural information flows is defined as anterograde, and the reverse direction is named retrograde. Therefore, retrograde viral tracers, spreading from post-synaptic neurons to pre-synaptic ones, are suitable to map input networks, while the output network should be mapped by anterograde tracers, which transmit in the opposite direction. Most transmittable viral tracers replicate in the initially infected neurons, and then the progeny viruses further invade the connected neurons.

During the past decades, non-transmittable retrograde tracers derived from the adeno-associated virus (AAV) [25,26,27], canine adenoviral vector (CAV) [28,29], and transmittable retrograde tracer derived from rabies virus (RABV) [30,31,32,33] have been well-developed and broadly applied. Certain strains of the α-herpes virus, such as pseudorabies virus (PRV) strain Bartha [34,35,36] and HSV-1 strain MacIntyre [37], were also shown to retrogradely transmit in CNS. Several reviews and articles have given systematic introductions, comparison, and instruction on these well-established retrograde tracers for different research purposes [38,39,40].

Compared to the well-developed retrograde tracers, the development of anterograde transneuronal tracers lags behind [41]. There are only a few choices available for anterograde transneuronal tracing, including AAV [42], vesicular stomatitis virus (VSV) [43,44], and HSV-1 strain H129 [45,46,47]. Although traditionally considered as a non-transmittable tracer, high concentrations of recombinant AAV1 and AAV9 were shown to be able to transport anterogradely in the axon and spread to the post-synaptic neurons, thus serving as an anterograde transneuronal tracer [42]. However, their transneuronal spreading efficiency is so low that the tracing result is invisible with fluorescent protein directly expressed by AAV. Therefore, a proper signal amplification strategy, such as Cre recombinase-dependent fluorescent reporter, is required to visualize the tracing result. Nevertheless, we need to be aware of the caveat that AAV-Cre is also able to retrogradely spread to presynaptic neurons, resulting in Cre-dependent transgene expression in presynaptic neurons. VSV pseudotyped with G protein of lymphocytic choriomeningitis virus (LMCV) was reported to spread anterogradely, but it was later found to be due to potential contamination [43]. Recently, Lin et al. reported VSV-NR7A, a recombinant VSV with N gene mutation, as an anterograde tracer with low toxicity [44], which still requires more time and data to confirm.

## 4. H129-Derived Anterograde Tracers

H129-derived tracers represent the most promising anterograde tracers so far [18,48,49]. Unlike most of the HSV-1 strains, which spread bidirectionally, the HSV-1 strain H129 shows a unique feature of predominant anterograde transmission. The mechanism of anterograde directional transportation and transmission remains unclear [50]. The low-passage clinical HSV-1 strain H129 was originally isolated from the brain of an acute necrotizing encephalitis patient in 1977 (Figure 1A) [51]. Zemanick et al. first reported the predominant anterograde transneuronal transmission of H129 when employing it on Cebus monkey (*Cebus apella*) for neuronal circuit tracing in 1991 [17]. After this unique feature was carefully verified in both rodent and non-human-primate, H129 was soon used to trace trigeminal afferent pathways and visual systems [17,47,52]. Subsequently, several groups applied the wildtype H129 as an anterograde tracer to map output circuits [45,46,53,54]. However, the requirement of immunostaining or in situ hybridization for signal visualization of the wildtype H129 limited its application (Table 1).

Modern genetic engineering techniques boost the development of H129-derived anterograde neuronal tracers. The first breakthrough was achieved in 2011 [48]. Lo et al. disrupted the original coding gene of the thymidine kinases (TK) (also known as *UL23*) with a bicistronic expression cassette (tdTomato-2A-TK), and placed the Lox-Stop-Lox element between the CAG promoter and the tdTomato-2A-TK cassette. In this way, the recombinant H129 H129dTK-TT was generated, which has been used as a Cre-dependent anterograde polysynaptic tracer (Figure 1B) [48]. In the neurons expressing Cre recombinase, the Stop signal in the Lox-Stop-Lox element is permanently removed by the Cre recombinase, resulting in TK expression and H129dTK-TT replication. Notably, the removal of the Stop signal by Cre recombinase is irreversible, so H129dTK-TT is transferred to a wildtype-like virus in Cre-expressing neurons, and it may further transmit anterogradely to downstream neurons. Therefore, H129dTK-TT can be used as an antegrade polysynaptic tracer to map the output pathways from a given genetically marked neuronal subpopulation with undiluted fluorescent reporters (Table 1). It has been applied by many laboratories worldwide and contributed to many achievements [55,56,57,58].

Soon after Lo developed H129dTK-TT, McGovern et al. also generated H129-EGFP by inserting a CMV promoter controlled EGFP expressing cassette between *UL26/26.5* and *UL27* (Figure 1C) [59]. Unlike H129dTK-TT, which can only start the anterograde transmission from the Cre-expressing neurons, H129-EGFP has no transmission limitation in the starting neurons. It may infect any neurons around the tracer injection site, replicate in them, transmit to the next order of neurons, and repeat the replication and transmission process to achieve polysynaptic tracing (Table 1). It has been used as the polysynaptic tracer for lateral septal nucleus projection [60].

Although H129dTK-TT and H129-EGFP both express fluorescent proteins, their labeling intensity remains relatively low, and often requires immunostaining against the fluorescent protein to amplify the signal. To overcome this, in 2017, our laboratory (Zeng et al.) generated H129-G4, which represents so far the brightest anterograde tracer [18]. H129-G4 contains 2 copies of the binary-GFP expression cassette, which is composed of the membrane-bound EGFP (mEGFP) and EGFP with 2A linker (Figure 1D). The total of 4×GFP copies greatly increased the expression level of fluorescent protein, and thus strengthens the labeling intensity. Similar to H129-EGFP developed by McGovern, H129-G4 starts the anterograde transneuronal transmission from any infected neurons without specificity (Table 1). The strong labeling intensity of H129-G4 makes it also capable of visualizing the detailed morphology of the labeled neurons, and thus it is compatible with the powerful imaging technique fMOST (fluorescence Micro-Optical Sectioning Tomography) [18,61]. So far, this is the only anterograde transneuronal tracer that can be used together with fMOST to decipher the whole-brain projection and the neuronal morphology of a given brain region. H129-G4 has been applied by several groups and contributed to many publications [62,63,64,65,66,67,68,69].

Recently, a novel H129-derived high-brightness anterograde polysynaptic tracer, H8, was introduced [70]. Su et al. utilized a different strategy to increase the fluorescent protein expression level via a “Trojan horse-like” strategy. Briefly, the AAV-ITR flanked GFP expressing cassette was inserted into the H129 genome together with the AAV *Rep* gene, generating H129-H8 (Figure 1E). After H129-H8 infects neurons, the H129 genome replicates and expresses GFP. Meanwhile, the ITR-flanked GFP expression cassette also replicates independently with the assistance of *Rep* gene products. The GFP expression from both H129 and AAV-ITR dramatically increases the GFP synthesis amount. Using H129 as a “Trojan horse” to hold the AAV-ITR flanked GFP expression cassette allows H129-H8 to have high labeling brightness (Table 1).

All of the above mentioned H129-derived tracers are replication-competent and may spread through neurons unlimitedly. They allow for the efficient mapping of the multi-order downstream targets as polysynaptic tracers (Figure 1H), but fail to determine whether the two brain regions are directly connected or mediated through other regions. Therefore, a tracer transmitting for only one order (monosynaptic) is required to meet this demand. The common strategy for monosynaptic tracing normally uses two recombinant viruses. The major one is the mutant viral tracer with an essential gene deficiency. And the helper virus, normally AAV, expresses the deficient gene complementarily to assist the replication of the major monosynaptic tracer (Figure 1I). RABV with G protein coding gene deletion (RVdG) has been well-developed and used as a retrograde monosynaptic tracer for over a decade [30,71]. However, the development of anterograde monosynaptic tracers, which are urgently needed, has fallen far behind.

The first milestone work was done by Zeng et al. in 2017 [18]. Besides the H129-G4, Zeng et al. also developed the first anterograde monosynaptic H129-dTK-tdT (Figure 1F). Using a strategy similar to Lo et al. to develop the H129dTK-TT, Zeng achieved monosynaptic tracing by controlling TK. TK is an essential enzyme in the thymidine triphosphate synthesis pathway [33], and thus plays a crucial role in viral genome DNA replication. TK deficiency severely impairs the replication of H129-dTK-tdT in neurons. The helper AAV-TK-GFP expresses TK in trans to assist H129-dTK-tdT replication, and the newly produced virus progenies anterogradely transmit to the innervated neurons downstream. There, the virus fails to replicate due to the absence of TK expressing AAV. Along with virus transmission, all H129-dTK-tdT infected neurons are labeled by tdTomato. However, the impaired virus replication in the downstream neuron also results in low tdTomato expression level. The labeling intensity of H129-dTK-tdT is too weak to be directly observed, so immunostaining is required to visualize the labeling signal (Table 1) [18]. By inserting the secondary tdTomato expression cassette into an optimized locus in H129-dTK-tdT genome between US7 and US8, we have generated the improved version H129-dTK-T2 with increased labeling intensity [49] (Figure 1G) (Table 1).

## 5. Limitations of Current H129-Derived Anterograde Tracers

### 5.1. Labeling Intensity and Distribution

Visualizing the neuronal circuit requires a high-intensity labeling signal. Simply inserting a fluorescent protein expression cassette into H129 genome failed to yield a high enough labeling intensity, possibly due to the limited fluorescent protein expression level. H129dTK-TT and H129-EGFP both showed insufficient labeling intensity, making it difficult to clearly visualize the labeled neurons, especially without immunostaining to amplify the signal (Figure 2A). Further developed polysynaptic tracers, H129-G4 and H129-H8, have achieved better labeling intensity via different strategies. However, the labeling signal of the monosynaptic tracers remains poor, especially in the post-synaptic neurons (Figure 2B). The overall morphology and details of the neuron structure cannot be well characterized and clearly observed by this insufficient labeling intensity. Besides intensity, the labeling signal requires even distribution from the soma throughout the axonal terminal. Thus, the overall morphology and details of the neuron structure can be well characterized and clearly observed. Neuron somas are usually about 10–25 μm in diameter, but the average diameter of axons is only about 1 μm. The huge size difference causes uneven fluorescence intensity between the soma and neurites (Figure 2C). When tracers appropriately label the neuron soma, less efficient labeling intensity is seen in the neurites. On the flipside, if axons are clearly visualized by sufficient fluorescent proteins, the soma will be overexposed. This is one of the largest challenges for the whole brain neuronal tracing via fMOST. The only available solution so far is sparse labeling, as introduced in multiple studies [72], but all current sparse labeling methods all in transneuronal labeling.

### 5.2. Potential Retrograde Labeling and Transmission

Besides predominantly invading somas, H129 is occasionally picked up by the axonal terminal of the upstream neurons. Then it is retrogradely transported to the soma by dynein and expresses fluorescent proteins (Figure 2D) [73]. H129 has also been shown to have delayed retrograde transmission and labeling (Figure 2E) [74,75], although this is still controversial. Both terminal pickup and retrograde transmission may result in non-specific labeling of the upstream neurons and mislead the analysis of the tracing results.

### 5.3. Transneuronal Transmission vs. Transsynaptic Transmission

Whether the tracers transmit exclusively through the synapse is a serious concern about most current transneuronal tracers. To reveal the neuronal circuit accurately, the applied tracers must transmit from one neuron to the connected ones only via synapses, so-called transsynaptic transmission. However, there is no direct solid evidence to prove the transsynaptic transmission of H129. The synapse gap is normally 20 nm, but the average diameter of H129 virion is about 200 nm. It is still a mystery whether and how such a large virus particle passes through such a small gap. Before the transsynaptic transmission of H129 is confirmed, is recommended to describe the H129-derived tracer as transneuronal rather than transsynaptic. HSV-1 has been shown to spread to adjacent cells through varicosity (Figure 2F) [76,77]. However, whether H129 also transmits to adjacent neurons through varicosity, which will be a large challenge for the accuracy of H129 antegrade tracing, remains to be revealed.

Neurons are not the only cell type in CNS infected by HSV-1. The fMOST images in Zeng’s publication showed that H129-G4 also infects astrocytes [18]. Although there is no solid evidence indicating H129 spreads from these infected astrocytes to adjacent neurons, this possibility still needs to be taken into consideration when analyzing H129 tracing results. The monosynaptic tracer H129-dTK-tdT shows impaired replication in neurons due to the lack of TK. However, the cellular thymidine kinases in astrocytes may compensate for the loss of viral TK, and assist H129-dTK-tdT replication. If H129-dTK-tdT infects astrocytes, it may replicate normally and possibly transmit to adjacent neurons after egressing or releasing from the initially infected astrocytes (Figure 2G). This will also cause non-specific tracing and mislead the result analysis. Although neither our lab nor other groups observed H129-dTK-tdT infecting astrocytes, this potential risk cannot be completely excluded. Besides H129-derived tracers, AAV, RABV, and PRV also face the same problem.

### 5.4. Toxicity to the Infected Neurons

Toxicity, including both cytotoxicity and animal toxicity, represents the major concern of all current transneuronal viral tracers. After infecting neurons, viral tracers must replicate to produce viral progenies, which further transmit to the connected neurons. The virus hijacks cellular machinery and alters regular cellular physiological processes for viral biosynthesis, which will severely damage the infected cells. The virus spreading and the following viral replication cause severe dysfunction or even death in large amounts of neurons (Figure 2H). In addition, the induced inflammatory responses also contribute to neuron death and brain tissue damage. Normally the polysynaptic tracer H129-G4 kills the experimental mice in 3–7 days, depending on virus dosage and the injected brain region [70]. Even if the animal remains alive, brain slices show severe inflammation and tissue damage around the injection site. Pathological phenotypes such as irregular neuron morphology, damaged axons, and abnormal axon swelling are observed in the labeled neurons. Cytotoxicity is reduced with the replication-impaired monosynaptic tracer H129-dTK-tdT, but only without complementary TK expression by the helper AAV. In the AAV-assisted H129-dTK-tdT replication starter neurons, severe damage is observed within 3–7 days, and starter neurons co-infected by H129-dTK and AAV-TK cannot be detected at 10 days (Figure 2I) [18].

### 5.5. Difficulties in H129 Genetic Manipulation

The large size genome of HSV-1 makes it impossible to be manipulated using regular molecular cloning strategies. The traditional method to generate recombinant HSV-1 is homologous recombination in eukaryotic cells. Briefly, the designed cassette flanked by the homology arms is transfected into HSV-1 infected cells, or co-transfected together with HSV-1 genome DNA. With successful recombination, the virus is then isolated by plaque purification according to fluorescence or other selection markers. The recombination in eukaryotic cells may cause unexpected mutations on the viral genome due to the host cell selection pressure [78,79]. The low recombination efficiency also makes it a time- and energy-consuming task to generate large amounts of various recombinant H129-derived tracers. H129dTK-TT, H129-EGFP, and H129-H8 were all generated using this method.

An alternative method based on the bacterial artificial chromosome (BAC) technique, greatly facilitates the construction of recombinant viruses. The H129 genome is integrated into the BAC sequence by homologous recombination in eukaryotic cells in the same way as described above. Then, the circular recombinant genome BAC-HSV1 is isolated from eukaryotic cells and transformed into *E.Coli*. Further genomic modification is performed by homologous recombination in *E.Coli*, which has high efficiency and fidelity, with the help of resistant selection marker(s). The recombinant virus is then reconstituted by transfecting the BAC-HSV1 DNA isolated from *E.Coli* into eukaryotic cells [78,79]. Using this BAC-based process, our laboratory constructed the toolbox of H129-derived tracers, including H129-G4, H129-R4 (similar to H129-G4 but with 4×mCherry), H129-dTK-tdT, H129-dTK-T2, and many other variants. However, the current methods by which H129-derived tracers were developed one-by-one is not suitable for generating a large number of recombinant virus variants that can be screened in a high throughput manner.

## 6. Strategies for Optimizing and Developing Future H129-Derived Tracers

To better characterize the output neuronal pathways, improved anterograde tracers with high labeling efficiency, intensity, specificity (direction and cell type), and low/non-toxicity are urgently required. Since the limitations of H129-derived tracers described above are all intrinsically related to the virological properties of HSV-1 or H129, novel strategies should be applied to modify H129 and develop improved anterograde tracers.

### 6.1. Increasing Labeling Intensity

Current viral tracers mostly use EGFP, mCherry, or tdTomato to label neurons. Choosing improved fluorescent proteins with higher quantum yield may achieve brighter labeling, and thus increase detection sensitivity and imaging quality. For example, RRvT is a tdTomato variant with a similar spectrum but a higher quantum yield [80]. Replacing tdTomato by RRvT in the same expression cassette should result in a 25% increase in the labeling intensity. Photostability and spectra, in addition to pure brightness, should be also taken into account to achieve stable, compatible, and bright labeling.

The expression level of the fluorescent protein represents another most important attributor for labeling intensity. Stronger promoters, more powerful enhancers, and optimized 3′/5′-UTRs in the expression cassette all contribute to boosting the fluorescent protein expression level. Different gene loci in HSV-1 genome display different expression levels, possibly caused by epigenetic modification [81]. Therefore, the insertion site of the fluorescent protein expression cassette in H129 genome should also be carefully chosen to reach maximum expression, especially for the replication-deficient monosynaptic tracers [82,83].

It has both shown to be a successful strategy to increase the expression cassette copy number as Zeng et al. did, and to amplify the expression cassette aside from the H129 genome replication as Su et al. did [18,70]. In addition, the self-amplified RNA (saRNA) system derived from an alpha-virus, such as the Sindbis virus (SINV), also provides another potential option [84]. By applying this strategy, the expression cassette of the fluorescent protein is cloned into the saRNA vector and integrated into the H129 genome. The expression cassette replicates both as DNA by the H129 replication system or as RNA by the SINV replication system. Transcription of the fluorescent protein occurs from all these replicated expression cassettes. Therefore, the expression level of the fluorescent protein should be dramatically increased. This is especially useful for long term expression of functional elements via the replication-deficient H129 tool [85,86].

One possible strategy to improve the evenness of the entire cell labeling intensity is to enhance the localization of the fluorescent protein in neurites. The neurite localization tag fused to the fluorescent protein is a potential solution. Kameda et al. reported myrGFP-LDLRct as an excellent synthetic protein for dendritic visualization [87] because of its high efficiency for targetting neurites. Replacing the original GFP with this myrGFP-LDLRct in H129 tracers should greatly boost the labeling intensity in neurites while maintaining acceptable soma labeling. This will benefit the morphological analysis of anterograde neuronal circuits.

### 6.2. Reducing Non-Specific Tracing

Virus entry to a defined cell type is the first step for establishing a successful infection, which is dependent on certain envelope glycoproteins for the enveloped virus (reviewed in [88,89]). RABV has only one glycoprotein (G). Replacing RV-G with EnvA on RVdG (G deleted RABV) envelope, generating RVdG-EnvA, switches the pseudotyped virus to exclusively infect cells with surface TVA [30]. This pseudotyped RVdG-EnvA allows retrograde circuit tracing from a defined TVA-expressing starter neuron population with reduced terminal invasion [71]. However, the 13 glycoproteins and complicated infection mechanisms of HSV-1 make it impossible to utilize the EnvA/TVA strategy.

“Retargeting” the H129-derived tracer might be one key strategy. Multiple glycoproteins are involved in the entry process of HSV-1. gH/gL heterodimer and gD are responsible for binding to the cell surface receptor, and gB induces the membrane fusion of the virion and cellular envelope [90]. By deleting gD and modifying the “core fusion machinery” gB/gH/gL, the infection tropism of HSV-1 can be switched to a specific cell type, also known as “retargeting”. Over the past two decades, great progress in HSV retargeting has been achieved for gene delivery and oncolysis (reviewed in References [91,92]). The structure characterization of HSV fusogen gB will further benefit the retargeting strategy [93]. For example, HER2 is a member of the human epidermal growth factor receptor family. It is highly expressed in certain human breast cancer cells. By loading the single-chain variable fragment (scFv) against HER2 to the gH/gL complex, Leoni et al. successfully retargeted HSV-scFv-HER2 to infect cells expressing HER2 [94]. The scFv/HER2 represents a potentially promising retargeting strategy for H129. Similar to EnvA/TVA for the RABV tracer, a retargeted H129 tracer in combination with AAV expressing HER2 (maybe only the extracellular domain to reduce the potential signaling influence to cells) may reduce non-specific tracing caused by terminal pick-up. Pre-administrating AAV-HER2 leads HER2 expression only in neurons around the injection site. Then H129-scFv-HER2 injected to the same site will only invade the cells expressing HER2. Therefore, the terminal pickup chance by neurons in other areas is dramatically decreased, resulting in reduced non-specific tracing. When using AAV with Cre-dependent HER2 expression in Cre transgenic animals, this strategy also allows anterograde tracing initiated from a defined (Cre+) neuron population. This will achieve a more precise neuronal circuit dissection and makes the applications more flexible [95]. Moreover, the high binding affinity of the scFv to HER2 might further enhance the virus invading the specific target cells, and thus increase infection and tracing efficiency.

Reducing the terminal pickup of H129-derived tracers is another key strategy. David et al. reported less terminal pickup of HSV-1 pseudotyped with a swapped gK from the HSV-1 KOS strain [96]. Our laboratory also confirmed this result and found that replacing H129-gK with KOS-gK reduces the terminal pickup by 90% (unpublished data). Although the exact mechanism remains to be revealed, H129 virus pseudotyped with KOS-gK should be an alternative strategy to reduce non-specific terminal labeling. The identification and modification of terminal pickup-related viral structural proteins will help to reduce terminal pickup and improve specific tracing.

Reducing the retrograde transmission of H129 is another key strategy. Although remaining controversial, the potential retrograde transmission of H129 represents an obstacle to rigorously elucidating neural circuits. Therefore, minimizing or even eliminating retrograde transmission is an important task for developing improved H129 tracers. The intercellular transmission and intracellular transportation of HSV-1 are complicated multistage processes that require several different cellular compartments and the coordination of many viral and cellular proteins [97]. HSV-1 axonal transport and transneuronal transmission have been reviewed in-depth [98,99,100,101]. It has been shown that the envelope protein pUS9 and the tegument protein pUS11 play major roles in HSV-1 anterograde transmission by interacting with the motor protein kinesin-1 [102,103]. Glycoprotein gE and gI are also reported to be required for anterograde transmission [104,105]. Different HSV-1 strains with different transmission properties, including F, 17, H129, and McIntyre-B, have been sequenced and compared for their transmission properties [106]. Dong et al. recently showed that H129 capsid and envelope are transported in the axon separately, and notably, capsids are enveloped at axonal varicosity and terminals [107]. However, the detailed mechanism of HSV-1 transmission, especially why H129 gains the preferential anterograde transmission, remains unclear.

Eliminating/reducing nonspecific retrograde transmission and increasing anterograde transmission efficiency are some of the key tasks for future H129 tracer development, which requires a better understanding of the underlying mechanisms.

### 6.3. Attenuating the Toxicity

The toxicity of H129-derived tracers is currently the biggest disadvantage and a problem that should be solved with the most urgency. Virus toxicity is mainly caused by viral virulence and inflammation; the key is the virulent viral proteins [108]. Therefore, modifying virulent viral proteins is an important strategy to reduce cytotoxicity.

Virus virulence-associated cytotoxicity is the main obstacle. By interacting with host proteins, the input and de novo synthesized viral proteins usually change the cellular environment and hijack the physiological process of the infected neurons to favor virus replication. This process also causes neuronal dysfunction or even cell death. Several proteins have been shown to be highly toxic, including but not limited to *ICP0* [109], *ICP4, ICP27* [110], *UL41* (vhs) [111] and *ICP34.5* (*γ34.5*) [112]. The most direct attenuation strategy is to delete these toxic viral genes. However, most of them are essential to virus replication, and their deletion will lead to replication deficiency. Therefore, this strategy is broadly applied to generate attenuated vaccines or gene delivery vectors [81,113], but not suitable for developing a transneuronal viral tracer. Currently, all viral vectors with non- or low-toxicity are replication-deficient and express no viral protein, such as AAV or lentivirus. However, replication-deficiency does not mean they are toxicity-free. Vectors derived from adenovirus (AdV) or SINV retain certain toxicity because they still express certain viral proteins. To label neuronal circuits, transneuronal tracers must replicate in the initially infected neurons, produce sufficient viral progenies, and then transmit to the neurons at the next order. According to the intrinsic virus property, virus replication is always accompanied by virus-caused cytotoxicity. Therefore, future H129 tracer development should aim at reaching a balance between virulence and tracing efficiency, obtaining low-virulent tracers rather than non-virulent tracers [114].

The toxicity of the RABV tracer has been reduced through G protein point mutations [115,116]. Compared to RABV (12 kb genome and 5 ORFs), H129 (152 kb genome and over 80 ORFs) is much more complicated. A recent study identified a total of 284 HSV-1 ORFs, highlighting a surprising complexity of HSV-1 gene expression [117]. Meanwhile, many confirmed virulence genes are indispensable for either replication or transmission. Our group has constructed a series of mutants with single/dual/multiple gene deletions, but none of them achieve the desired attenuation. Instead of deleting these toxic essential genes, a potential approach is to modify them to reduce their cytopathic effect while still retaining their proper function in viral replication. Deleting/modifying a series of non-essential toxic viral genes is an alternative option, but all these require an intensive understanding of the H129 viral gene function and protein structure.

Another strategy is to develop a conditional replicating H129 viral tracer. Ciabatti et al. generated the self-inactivating rabies virus (SiR) by adding a reversible destabilizing tag to the N protein, which allows the virus to undergo limited rounds of replication, which limits the level of virulent viral proteins and the resulting adverse effect [118]. Although the siR was later suspected to be just the first generation RVdG tracer (siR losing the modification tag on N protein), this “self-inactivating” strategy might be adopted to develop a conditional replicating H129 tracer by adding the PEST tag to an immediate-early essential gene.

In general, multiple strategies should be combined together to achieve a balance between replication/transmission efficiency and cytotoxicity. The attenuated anterograde tracer should express a high level of fluorescent proteins with the least necessary level of toxic viral gene expression, and maintain efficient anterograde transmission efficiency.

### 6.4. Developing High Throughput Modification and Screening Methods

Compared to traditional homologous recombination in eukaryotic cells, BAC-based techniques greatly simplify the genomic manipulation process and increase the efficiency of generating recombinant H129. But it is still time- and energy-consuming to pre-design the modification strategy and test the generated recombinant virus tools one-by-one. Directional evolution may represent a high throughput method to generate and screen novel H129-derived tools. This method produces a large number of random recombinant virus mutants, enriches meaningful mutations through selective pressure, and ultimately yields the desired phenotype and genotype. This directional evolution has achieved great improvements in AAV development by generating several useful artificial AAV serotypes [119]. However, directional evolution has not yet been applied to H129 tracer development. CRISPR-Cas9 techniques in combination with sgRNA libraries might provide a potential high throughput method to generate a large amount of recombinant H129 mutants, but it is still a challenge to develop the appropriate selection pressure and screening method.

## 7. Conclusions

Neurotropic H129 represents one of the most promising anterograde neuronal circuit tracers due to its predominant anterograde transmission and large transgene capacity. An H129-derived toolbox has been developed and has contributed to revealing many novel output neuronal pathways. However, current H129-derived tracers are far from being perfect anterograde tracers. The intrinsic limitations are closely related to their viral machinery and characteristics. These can only be overcome through virus modification after a thorough understanding of the virus itself. Two fundamental virological puzzles of HSV need to be urgently addressed in order to develop improved H129 tracers. The first is the mechanism of HSV-1 intracellular transportation and intercellular transneuronal transmission in CNS. The second is to annotate all functions of HSV-1 genes and proteins.

It is important to examine the advantages and disadvantages of current H129-derived tracers for further development and refinement of improved H129 anterograde tracers. Such refinements are expected to come both from the efforts to modify the H129 genome and from ongoing research characterizing HSV virology. Merging the findings from these two fields will yield a broader range of tracer platforms that can better serve mesoscale connectomics.

## Figures and Tables

**Figure 1 ijms-21-05937-f001:**
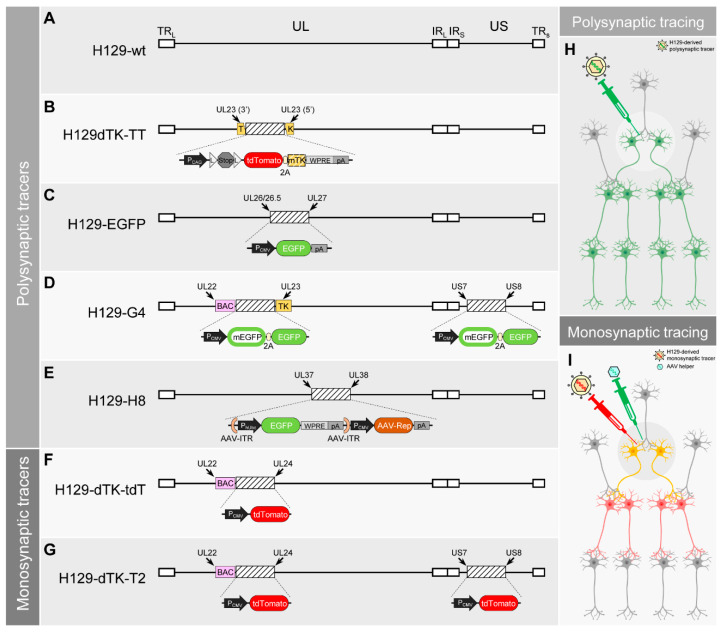
Genomes of current H129-derived tracers, and anterograde tracing schematics. (**A**–**E**). Genome schematics of current H129-derived polysynaptic tracers. (**A**) Wildtype H129 (H129-wt). The genome of HSV-1 strain H129 (H129) is composed of 2 regions, unique long (UL) and unique short (US), which are flanked by terminal repeat (TR) and internal repeat (IR), respectively. (**B**) H129dTK-TT. The coding gene of tdTomato and codon-modified *TK* (*mTK*) is linked by the sequence encoding 2A self-cleaving peptide (2A). A LoxP-Stop-LoxP element is placed between CAG promoter (P_CAG_) and tdTomato-2A-TK cassette. WPRE is used as an enhancer, and poly(A) (pA) is used to stop the transcription. The whole expression cassette is inserted into the middle of the *TK* gene (*UL23*) in the H129 genome. The original TK is knocked out by being split into 3′- and 5′- parts. (**C**) H129-EGFP. The EGFP expression cassette with CMV promoter (P_CMV_) and poly(A) is inserted into the H129 genome between *UL26/26.5* and *UL27*. (**D**) H129-G4. The BAC sequence is inserted into the H129 genome between *UL22* and *UL23* to generate BAC-H129. The P_CMV_ controlled binary-GFP expression cassette is composed of a membrane-bound EGFP (mEGFP) and an EGFP with a 2A linker. Two copies of the binary-GFP expression cassette are inserted into the H129 genome. One is placed between the BAC sequence and *UL23*, and the other between *US7* and *US8*. (**E**) H129-H8. The GFP expression cassette is composed of EGFP controlled by human ubiquitin C gene promoter (P_hUbc_) and WPRE-pA, which are all flanked by AAV2-ITR. The AAV replicase expression cassette is composed of the AAV2 *Rep* gene (*AAV-Rep*) and poly(A) (pA) controlled by P_CMV_. Both cassettes are inserted into the H129 genome between *UL37* and *UL38*. (**F**,**G**). Genome schematic of current H129-derived monosynaptic tracers. (**F**) H129-dTK-tdT. The tdTomato expression cassette driven by P_CMV_ is inserted into BAC-H129 to replace the *TK* gene (*UL23*), resulting in TK deletion. (**G**) H129-dTK-T2. Another identical tdTomato expression cassette (P_CMV_-tdT) is inserted into the genome of H129-dTK-tdT, between *US7* and *US8*. (**H**,**I**). Schematic diagram of polysynaptic and monosynaptic tracing. (**H**) Polysynaptic tracing. The H129 polysynaptic tracers (indicated by the enveloped virion with green genome) are replication-competent. After being injected into the brain region (indicated by syringe and white circle), they infect the local neurons, replicate, and express fluorescent proteins to label the neurons (indicated as green). The produced progeny virions transmit anterogradely to the next order downstream neurons, repeat the replication/transmission processes, and label further downstream neurons. The ideal polysynaptic tracers should not label upstream neurons (indicated as gray) via terminal pickup or retrograde transmission. (**I**) Monosynaptic tracing. The H129 monosynaptic tracers (indicated by the enveloped virion with red genome) are replication-incompetent due to certain gene deletion. The helper virus, mostly AAV (indicated by the unenveloped virion with green genome), expresses the deficient gene and the fluorescent protein of a different color (indicated as green). After being injected into the same brain region (indicated by syringes and gray circle), the helper virus expresses the gene to complimentarily support the replication of deficient H129 monosynaptic tracer in trans. The produced progeny virions transmit anterogradely to the next order neurons, where there is no helper virus. Then the monosynaptic tracers stop transmitting to the further downstream regions. During the monosynaptic transmission, the monosynaptic tracers label the neurons by expressing the fluorescent protein (indicated as red). Therefore, the initial coinfected neurons (starter neurons, indicated as yellow) are labeled by both H129 tracer (red) and helper (green), and the second-order neurons are labeled only by H129 tracers (indicated as red). No further downstream neurons or upstream neurons were labeled (indicated as gray).

**Figure 2 ijms-21-05937-f002:**
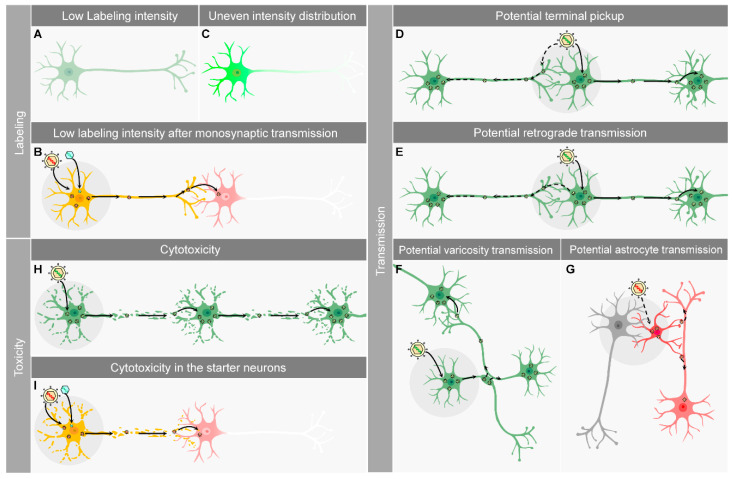
The limitations of current H129-derived tracers. (**A**–**C**). Limitations of labeling intensity. (**A**) Some tracers label the neurons with relatively low intensity. (**B**) Current monosynaptic tracers do not replicate after monosynaptically transmitting to postsynaptic neurons. Labeling intensity in second-order neurons is very low, making the neurites invisible even after immunostaining. (**C**) The labeling intensity of the entire neuron is not evenly distributed. The soma is labeled much more brightly than the neurites, resulting in axon and axonal terminals that are difficult to observe or even invisible. (**D**–**G**). Limitations of the tracer transmission. (**D**) H129-derived tracers mainly invade the neuron from the soma and transmit further (solid arrow). However, they can also be potentially picked up by the axonal terminal of the upstream neurons (dashed arrows), and label them with the fluorescent protein. (**E**) Besides the predominant anterograde transmission (solid arrows), H129-derived tracers were reported to potentially transmit retrogradely and label upstream neurons (dashed arrows). (**F**) H129 may potentially transmit to adjacent neurons from varicosity (indicated as the enlarged region of the axon) (dashed arrows). (**G**) H129 may potentially infect astrocytes. In astrocytes, H129-dTK may replicate in the absence of helper virus, and the progeny virions may transmit to adjacent neurons (dashed arrows). (**H**,**I**). Limitations of cytotoxicity. (**H**) The polysynaptic tracers (indicated by the enveloped virion with green genome) are replication-competent. Their anterograde transmission requires tracer replication and progeny production. The viral replication causes severe damage or even neuron death (indicated by the cracks). (**I**) Monosynaptic tracers (indicated by the enveloped virion with red genome) are replication-deficient in the absence of a helper virus. Due to replication deficiency, only a few viral proteins are synthesized, leading to less damage to the cells. Therefore, they show attenuated toxicity to the postsynaptic neurons after monosynaptic transmission (the red neuron). However, under the assistance of the helper virus (indicated by the unenveloped virion with green genome), H129-derived monosynaptic tracers replicate in the starter neurons (the yellow neuron) and cause severe damage to the starter neurons (indicated by the cracks).

**Table 1 ijms-21-05937-t001:** Current H129-derived anterograde transneuronal tracers and their properties.

	H129-wt	H129dTK-TT	H129-EGFP	H129-G4	H129-H8	H129-dTK-tdT	H129-dTK-T2
Polysynaptic Tracing ^a^	✓	✓	✓	✓	✓	✗	✗
Monosynaptic Tracing ^a^	✗	✗	✗	✗	✗	✓	✓
Labeling Brightness ^b^	-	+/++	++	+++	++/+++	+	+
Starter Neuron Specificity ^a^	✗	Cre+ neuron	✗	✗	✗	Naïve/Cre+/Flp+/… (controlled by helper AAV expressing TK)	Naïve/Cre+/Flp+/… (controlled by helper AAV expressing TK)
Advantages	Works in primates	Polysynaptic tracing from Cre+ neurons	Increased labeling intensity	With the brightest labeling so far	Enhanced brightness	Monosynaptic tracer, suitable for starter neuron specific or nonspecific tracing	Monosynaptic tracer, suitable for starter neuron specific or nonspecific tracing, increased labeling intensity
Limitations	No fluorescence, requires immunostaining, potential retrograde labeling, high toxicity	Low labeling intensity, can’t trace from naïve neurons, potential retrograde labeling, high toxicity	No starter cell specificity, potential retrograde labeling, high toxicity	No starter cell specificity, potential retrograde labeling, high toxicity	No starter cell specificity	Low labeling intensity, requires immunostaining to visualize post-synaptic neurons, potential retrograde labeling, relatively high toxicity in the starter neuron	Relatively low labeling intensity, potential retrograde labeling, relatively low toxicity in the postsynaptic neurons but still high in the starter neurons
Original Articles	[17,51]	[48]	[59]	[18,49]	[70]	[18,49]	[49]
Application Articles ^c^	[45,46,47,52,53,54]	[55,56,57,58] ^*^	[60]	[61,62,63,64,65,66,67,68]	/	[64]	/

a: ✓ Working; ✗ not working. b: - no fluorescence labeling; + weak labeling intensity; ++ moderate labeling intensity; +++ strong labeling intensity. c: * Not a full reference list, only representative articles are listed. / No published articles using these tracers so far.

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
