# Peer review of "Anterograde Neuronal Circuit Tracers Derived from Herpes Simplex Virus 1: Development, Application, and Perspectives"

_ijms, 2020, doi:10.3390/ijms21165937_

Round 1

Reviewer 1 Report

This is a well written review article titled Anterograde neuronal circuit tracers derived from Herpes simplex virus 1: development, application and perspectives.  Dong Li et al provide a detailed literature review that surveys HSV, and particularly the strain H129 of HSV as an anterograde tracing vector that can be utilized for biological evaluations of sensory neurons.  The article is comprehensive, and reviews other vectors, including AAV, sharing pros and cons of each vector described.  It is of use to the scientific community as a review.

Author Response

We would like to thank the Reviewer for their enthusiasm and positive feedback.  Please see the authors' response and detailed point-by-point reply in the attached document. 

Reviewer 2 Report

In this review, Li et al summarize the use of herpes simplex virus (HSV) strain H129 as a transneuronal tracer to analyze neuronal circuits. The review first introduces HSV and its neurotropism. After mentioning the traditional tracers, the authors describe viral tracers before focusing on HSV strain H129. This strain has the property of being anterograde-specific when it comes to trace neuronal circuits. A set of different H129-based constructs are depicted, and their advantages and limitations are well described in the text, figures and table. Finally, the authors concentrate on describing improvement needed to overcome limitations of H129-derived tracers. The manuscript is well organized, and the figures are clear. The references are appropriate. The text is nicely written but will benefit from a thorough review for English grammar, which could be improved.

Here are a few specific comments which the authors may consider in order to clarify some statements:

1- Although briefly alluded to (line 442), the molecular characteristics that make H129 specifically spread in an anterograde fashion are not described. What is known about these properties, specific to H129 should be mentioned when this viral strain is first described. In other words, describe what makes H129 a better than other strains for anterograde tracing.

2- Lines 100-106, 113-114: it seems that according to the authors definition, AAV should not be called “tracers” if they cannot be transmitted between neurons and therefore cannot trace circuits. It is also confusing when on line 121-123, the authors now suggest that AAV is an anterograde tracer. This suggests that AAV can transmit in an anterograde fashion, which contradicts the statements made before and undermines the use of AAV in monosynaptic tracing approaches. The description of AAV properties must be clarified to avoid contradictions and confusion.

3- It is mentioned that the reason why H129-dTK-tdT and H129-dTK-T2 can be monosynaptic tracers is their lack of TK gene, which can be complemented by AAV. H129dTK-TT also lacks TK. However, it is not considered suitable for monosynaptic tracing (table 1). Could it be conceivable that this virus could be used for monosynaptic tracing in combination, if so, would it be limited to Cre systems?

Other editorial points to consider:

4- Make sure there is a space before brackets containing reference numbers.

5- Sections 2 and 3 have the same title.

6- legend of figure 2 is labelled “figure 1”

7- Table 1: please explain the differences between “x”, “-“ and “/” . The meaning of these symbols needs to be clarified in table footnotes.

8- Lines 531-537 contain information on manuscript preparation and should be removed from the manuscript.

9- All reference formatting must be checked for consistency. For instance, capitalization of words in article titles is not consistent. Compare refs 1 and 13 with refs 3 and 14, for instance.

10- The correct citation for Reference 47 is:      

Beier, K. T., Saunders, A., Oldenburg, I. A., Miyamichi, K., Akhtar, N., Luo, L., Whelan, S. P. J., Sabatini, B., & Cepko, C. L. (2012). Erratum: Anterograde or retrograde transsynaptic labeling of CNS neurons with vesicular stomatitis virus vectors (Proceedings of the National Academy of Sciences (2011) 108, 37 (15414-15419) DOI: 10.1073/pnas.1110854108). Proceedings of the National Academy of Sciences of the United States of America, 109(23). doi.org/10.1073/pnas.1207087109

11- Some additional typographic and editorial errors. [Note that this is not a complete list. A thorough revision of the English grammar should still be undertaken.]

Lines 26, 73: the authors most likely meant “retain” and not “remain”.

Line 36: to the alpha-subfamily

Line 38: neurons

Line 55: use “latent” instead of “dormant”

Line 57: tissues

Line 60: remove “even if possible” or explain what is meant by this phrase.

Line 61: remove “it”

Line 95: awkward sentence. Needs to be rephrased

Line 99: powerful tracing tools in neuroscience research

Line 148: has been applied

Line 193: “TK is an essential enzyme in thymidine triphosphate.” This sentence is incomplete.

Line 233: “The idea polysynaptic tracers should not label upstream neurons (indicated as gray) by terminal pickup or retrograde transmission.” This sentence is unclear. Remove “the idea”?

Line 239: “complementary”, not “complimentary”

Line 468: shown to be highly toxic

Line 474: “viral” not “vial”

Line 517: replace “the most”  by “a very”

Author Response

We would like to thank the Reviewer for their positive feedback and excellent suggestions, which we have used to improve our article.  Following the suggestion of “The text is nicely written but will benefit from a thorough review for English grammar, which could be improved.”, Dr. Xu and his native English-speaking staff have provided English editing to correct sentence and grammar issues to improve the text readability.  

Please find the authors' response and detailed point-by-point reply in the attached document. 
